# CSDR Coupling with Exo III for Ultrasensitive Electrochemistry Determination of miR-145

**DOI:** 10.3390/molecules28052208

**Published:** 2023-02-27

**Authors:** Moli Zhang, Yang Yang, Lingyi Xin, Hua Zhang, Lun Wu, Jun Zhu, Jing Zhu, Shiyun Liu, Zhaohui Wang, Qinhua Chen, Guangyi Yang

**Affiliations:** 1Shenzhen Bao’an Authentic TCM Therapy Hospital, Shenzhen 518102, China; 2Shenzhen Bao’an Traditional Chinese Medicine Hospital, Guangzhou University of Chinese Medicine, Shenzhen 518000, China; 3Sinopharm Dongfeng General Hospital, Hubei University of Medicine, Shiyan 430345, China

**Keywords:** electrochemistry, miRNA-145, stroke, CSDR, Exo III, MNPs

## Abstract

Recently, miRNAs have become a promising biomarker for disease diagnostics. miRNA-145 is closely related to strokes. The accuracy determination of miRNA-145 (miR-145) in stroke patients still remains challenging due to its heterogeneity and low abundance, as well as the complexity of the blood matrix. In this work, we developed a novel electrochemical miRNA-145 biosensor via subtly coupling the cascade strand displacement reaction (CSDR), exonuclease III (Exo III), and magnetic nanoparticles (MNPs). The developed electrochemical biosensor can quantitatively detect miRNA-145 ranging from 1 × 10^2^ to 1 × 10^6^ aM with a detection limit as low down as 100 aM. This biosensor also exhibits excellent specificity to distinguish similar miRNA sequences even with single-base differences. It has been successfully applied to distinguish healthy people from stroke patients. The results of this biosensor are consistent with the results of the reverse transcription quantitative polymerase chain reaction (RT-qPCR). The proposed electrochemical biosensor has great potential applications for biomedical research on and clinical diagnosis of strokes.

## 1. Introduction

Stroke is a devastating cerebrovascular disease characterized by an interruption of the blood supply to the brain leading to neurological dysfunction, which can be broadly classified into an ischemic stroke or a hemorrhagic stroke [1]. Stroke is the fourth leading cause of death worldwide, resulting in 5.5 million deaths per year [2]. The quality of life of stroke survivors and their families is also severely affected [3]. There is a lack of effective treatment modalities to fully rehabilitate stroke patients. The optimum solution is to find the high-risk population of stroke early and give timely intervention treatment [4]. At present, the relevant techniques for the early diagnosis of stroke mainly include magnetic resonance imaging (MRI) and carotid imaging [5,6]. However, the above early diagnostic modalities need to be performed after lesion formation. Liquid biopsy is an easy way which has been widely used for the early diagnosis of many diseases [7]. The performance of liquid biopsy technology mainly depends on the two vital factors: disease-related biomarkers and the corresponding detection platform [8]. Studies have shown that the expression of miRNA-145(miR-145) in the peripheral blood of stroke patients is significantly higher than that of healthy people. This indicates miR-145 can be used as a potential biomarker for the early diagnosis of stroke.

The traditional methods for detecting miRNA mainly include the reverse transcription quantitative polymerase chain reaction (RT-qPCR) and Northern blotting, which are considered to be the gold standard in many applications [9,10]. However, the sophisticated instruments required, the time-consuming work involved, and the high cost of these methods limit their wider application. Therefore, there is an urgent need to develop an easy and cost-effective miR-145 detection method for the early diagnosis of stroke. With the advantages of facile operation, low cost, and favorable flexibility, biosensors such as nanopores, field effect transistors (FET) [11], fluorescence [12], and surface-enhanced Raman spectroscopy (SERS) [13] have attracted extensive attention for the sensing of miRNA in recent years. Based on its portability, easiness to integrate, and low cost, electrochemistry stands out from these biosensors for the detection of miRNA. Electrochemistry biosensors still pose some challenges to detecting miR-145 directly, mainly due to the low content, short length, and high sequence homology of miR-145 in the blood [14]. Hence, electrochemistry biosensors need to be combined with some strategies which could effectively convert the target miR-145 into an amplifying electrochemical signal for improving the sensitivity and specificity.

Various signal amplification strategies including rolling circle amplification (RCA) [15], CRISPR/Cas13a, duplex-specific nuclease (DSN) [16], and hybridization chain reactions (HCRs) [17] have been coupled with electrical biosensors for the biosensing of miRNA. The cascade strand displacement reaction (CSDR) [18], as an entropy-driven reaction, has attracted wide attention for its facile operation, high efficiency, and low cost. The CSDR can transform one miRNA into plenty of pieces of DNA, but its limited signal amplification capability still makes it difficult for sensing miR-145 with extremely low abundance. The combination of two different signal amplification strategies is an effective solution to overcome this. We successfully coupled the CSDR with CRISPR/Cas13a for ultrasensitive fluorescence determination of exosomal miRNA-21. Exonuclease III (Exo III) as a sequence-independent nuclease can digest DNA with a flat or blunt 3’-OH in double-stranded DNA [19]. Its enzymatic activity is highly structurally dependent. Hence, the CSDR’s coupling with Exo III will be a promising strategy to achieve the ultrasensitive detection of miRNA-145. It is regrettable that there are no reports on the combination of CSDR and Exo III for the electrochemistry determination of miRNA. In the CSDR, products and reactants co-existing in the same solution will result in a high background signal, which is unbeneficial for improving the sensitivity of biosensors. Magnetic nanoparticles (MNPs) with excellent separation properties are an ideal tool to eliminate substrate interference [20].

Based on the abovementioned factors, we designed an electrochemical biosensor via coupling the CSDR with Exo III and MNPs for the ultra-sensitive determination of miR-145. miR-145 can trigger the CSDR via an entropy-driven reaction resulting in a large amount of probe 3 (P3) being released from the MNPs. After magnetic separation, the free P3 can specifically hybridize with the methylene-blue-modified probe 5 (MB-P5) which was assembled on the electrode’s surface. The MB-P5 in the complex of MB-P5/P3 can be digested by Exo III resulting in a decrease in the electrochemical signal. Therefore, we can realize the ultrasensitive and special detection of miRNA-145 via the change in electrochemical signal. This developed electrochemistry biosensor will offer a new avenue for the early diagnosis of stroke.

## 2. Results and Discussions

### 2.1. The Principle and Feasibility of This Biosensor

Figure 1 illustrates the principle of this developed electrochemistry biosensor. First, biotin-modified P1 was hybridized with P2 and P3 to form the complex of P3/P2/P1; the formed complex combines with the SA-MNPs via the specific binding of biotin with SA. After the magnet is separated, the complex of P3/P2/P1/MNPs was obtained. Then, the testing sample and P4 were mixed with the complex of P3/P2/P1/MNPs. If the testing sample contained miR-145, miR-145 can trigger the toehold-mediated SDR via entropy drive to form the complex of miR-145/P2/P1/MNPs and release the free P3. The complex of miR-145/P2/P1/MNPs contains the new toehold resulting in the P4 triggering the SDR to release the free miR-145. The free miR-145 will initiate a new round of SDR, leading to the formation of new free P3. Thence, one target miR-145 can transform to the formation of numerous free P3s via the CSDR. After magnetic separation, the supernatant containing the free P3 was collected. The P5-modified GE was then immersed into the collected supernatant along with the addition of Exo III. The P3 hybridizes with the P5 to form the complex of P3/P5 on the surface of GE. Exo III can digest the ssDNA with a blunt or recessed 3-terminus in double-stranded DNA. In the complex of P3/P5, the 3-hydroxyl end of P5 has a recessed structure. Thence, the 3-terminus of P5 is continuously digested by Exo III, leading to the release of P3. The release of P3 will trigger a new cycle of hybridization and Exo III digestion. One P3 can initiate a large number of MB-modified P5s being digested via such a cyclic binding hydrolysis process, leading to a drastic decrease in the electrochemical signal. The detection event of miR-145 can be efficiently and specifically transformed to the change of measurable electrochemical signal via the developed electrochemical biosensor based on the coupling of CSDR and Exo III.

In order to verify the feasibility of the developed electrochemical biosensor for the detection of miR-145, a series of experiments were carried out. Electrochemical impedance spectroscopy (EIS) and cyclic voltammetry (CV) were used to evaluate the electrochemical performance of the modified electrodes at three different stages. As shown in Figure 1A, the CV results of the modified electrodes at various stages are given (5 mM potassium ferricyanide in 0.1 M KCl). The bare GE exhibits the maximum redox current (Figure 1A(a)). Meanwhile, the oxidation peak potential has the minimum difference from the reduction peak potential. The bare GE exhibits the best conductivity. The GE modified with the P5 has the minimum redox current as well as the maximum redox peak potential difference (Figure 1A(b)). This is mainly ascribed to the fact that the electrostatic repulsion between DNA and potassium ferricyanide was not conducive to the arrival of potassium ferricyanide onto the electrode surface. After the P5-modified GE is used for detecting miR-145, the redox peak current of the GE is larger than that of the P5-modified GC (Figure 1A(c)). This is due to the fact that the P5 modified on the electrode was digested by Exo III, leading to a decrease in the number of nucleotide bases on the GE that are beneficial for the electron transport of potassium ferricyanide. As shown in Figure 1B, the bare GE has the smallest semicircle diameter in the Nyquist diagrams (Figure 1B(a), ~250 Ω). The semicircle diameter of the P5-modified GC in the Nyquist diagrams is about 3500 Ω (Figure 1B(b)). After being digested by Exo III, the semicircle diameter in the Nyquist diagrams is about 3000 Ω (Figure 1B(c)). The semicircle diameter represents the electron transfer resistance. The results of EIS are consistent with the results of CV which both indicate P5 successfully self-assembling on the GE, as well as the P5 being successfully digested by Exo III after being used for detecting miR-145.

### 2.2. Optimization of the Assay Conditions

To obtain the optimal performance for sensing miR-145, we investigated the effects of various, mainly experimental conditions. The △i obtained via DPV was used as the main indicator to evaluate the performance of the biosensor. The amount of MNPs was firstly investigated. As shown in Figure 2A, the △i becomes larger with the increasing number of MNPs. When the MNPs exceed 20 μg/mL, the value of △i reaches the plateau. Too low of a concentration of MNPs can lead to too crowded DNA structures on the surface of the MNPs. This is very unbeneficial for the CSDR due to the spatial hinder. Twenty μg/mL was used as the optimal number of the MNPs. The △i at various concentrations of P1 was studied (Figure 2B). In the range from 10 nM to 30 nM, the △i was positively correlated with the concentration of P1. When the P1 was more than 30 nM, the △i decreased with the increasing concentration of P1. The high concentration of P1 implies the high density of DNA on the surface of the MNPs, leading to the inefficiency of CSDR. Thence, 30 nM was set as the optimal concentration of P1. Meanwhile, P2 and P3 had the same concentration as P1. As shown in Figure 2C, the △i keeps increasing as the CSDR time increases from 10 min to 40 min. When the CSDR time exceeds 30 min, the change in △i is very small. To obtain a good performance in a shorter amount of time, 30 min was used as the optimal time of the CSDR. The concentration of P5 affected the value of △i from two aspects. The higher of the concentration of P5, the greater the I_0_ was. However, too high of a concentration of P5 will lead to a too high density of DNA on the electrode’s surface, which is not conducive to the hybridization of P5 with P3. Two uM were used as the optimal concentration of P5 (Figure 2D). As shown in Figure 2E,F, as both the amount of Exo III and the time of Exo III digestion increased, the △i became larger. When the Exo III concentration and the Exo III digestion time exceeded 2.0 U/mL and 30 min, respectively, the △i for both reached a plateau. Therefore, the △i of the background always increased with the increase in the Exo III concentration and the Exo III digestion time. Thence, 2.0 U/mL and 30 min were used as the optimal Exo III concentration and the optimal Exo III digestion time, respectively.

### 2.3. Sensitivity and Specificity for the Detection of miR-145

Under the above optimal experimental conditions, the sensitivity and specificity of the developed electrochemical biosensor for the detection of miR-145 were investigated. Firstly, a series of various concentrations of miR-145 ranging from 0 to 1 × 10^7^ aM were detected. The corresponding DPV response results are shown in Figure 3A. The current signal intensity of DPV(I_1_) gradually decreased with the increasing concentration of miR-145. Moreover, the △i exhibited a good linear correlation with the common logarithm of the miR-145 concentration within the range from 1 × 10^2^ to 1 × 10^6^ aM (Figure 3B). The corresponding linear equation is △i = 0.3819*lgC − 0.0577 (R^2^ = 0.9957). C is the concentration of miR-145. The limit of detection of this electrochemical biosensor for the detection of miR-145 is about 100 aM. The performance of this work compared with the other published biosensors for detecting miR-145 is given (Table 1). We can see that the sensitivity of this developed biosensor is superior to most of the published miR-145 biosensors. The excellent sensitivity of this electrochemical biosensor for detecting miR-145 is mainly ascribed to the synergy effect of the following factors. The CSDR is capable of efficiently converting one miRNA into many DNA molecules, generating the “one to numerous” signal amplification. When the enzymatic activity of Exo III was triggered by DNA, the MB-labeled DNA modified on the electrode was cleaved, realizing the event of “detection of target” and transforming into the measurable change of electrochemical signal. Due to the powerful enzymatic activity of Exo III, one DNA trigger can lead to numerous MB-labeled DNA strands modified on the electrode, being cleaved, and thus generating the “one to numerous” signal amplification once again. The cascade signal amplification of the CSDR and Exo III can greatly guarantee the sensitivity of this biosensor. In addition, the MNPs can separate the reactants with products, thus eliminating potential interference from the reactants, reducing the blank signal and further improving sensitivity. The subtle combination of CSDR, Exo III, and MNPs can greatly improve the sensitivity of this biosensor for the detection of miRNA.

The specificity of the developed electrochemical biosensors has also been investigated. Several different mismatched RNAs from miR-21 such as single-base mismatch (SM, 10 pM), double-base mismatch (DM, 10 pM), triple-base mismatch (TM, 10 pM), and random sequence (SM, 10 pM) are all detected with the developed electrochemical biosensor. As shown in Figure 4, the △i of TM and RS was almost indistinguishable from the △i of the blank. The △i of DM is just slightly higher than the △i of blank. The △i of SM is significantly larger than the △i of blank, but the △i of SM is still much smaller than the △i of miR-145, even if the concentration of SM is ten times that of miR-145. These results all indicate that the fabricated electrochemical biosensor had excellent specificity toward the detection of miR-145. The superior specificity of the electrochemical biosensor for detecting miR-145 is mainly attributed to the synergy effect of the CSDR and MNPs. SM RNA initiation of the CSDR is much less efficient than miR-145 because the CSDR is specifically triggered by miR-145 in this electrochemical biosensor. In addition, the powerful separation of MNPs can reduce the potential interference of reactants which is very beneficial for the further improvement of specificity. Compared with other related techniques for the early diagnosis of stroke, the highly-developed, sensitive electrochemical biosensor in this study has the advantages of being easy to operate, cost-effective, and having low environmental and equipment requirements, and it has the potential for widespread application.

### 2.4. Clinical Samples

The blood samples of 10 healthy people and 10 stroke patients were used as real biological samples to verify the potential of this developed electrochemical biosensor for clinical application. The extracted total RNA was detected with this developed electrochemical biosensor. As shown in Figure 4A, miR-145 was of higher expression in stroke patients than in healthy people. Moreover, miR-145 expression in stroke patients and healthy people has significant statistical differences. This indicates that miR-145 has the potential as a biomarker for the diagnosis of stroke. In addition, the same samples were measured with the commercial quantitative reverse transcriptase polymerase chain reaction (qRT-PCR). We can observe that the results exhibited high accordance between the electrochemical biosensor and the qRT-PCR (Figure 4B). These results indicate that the electrochemical biosensor has good potential application for the clinical diagnosis of stroke.

## 3. Experimental Procedure

### 3.1. Materials and Reagents

All of the nucleic acid sequences used in this work were supplied by Sangon Biotech Co., Ltd. (Shanghai, China) and are listed in Table 2. Streptavidin-modified MNPs (SA-MNPs, 40 nm) were purchased from XFNANO Co., Ltd. (Nanjing, China, www.xfnano.com, accessed on 2 November 2020.). Exonuclease III was obtained from Takara Biotechnology Co. Ltd. (Dalian, China, www.takara.com.cn, accessed on 10 November 2020). 6-mercapto-1-hexanol (MCH) and tris (2-carboxyethyl) phosphine hydrochloride (TCEP) were purchased from Sigma-Aldrich (St. Louis, MO, USA). The other reagents used were of analytical grade and obtained from Aladdin (Shanghai, China). Ultrapure water obtained from a Millipore water purification system (18.2 MΩ·cm resistivity, Milli-Q Direct 8) was used in all of the runs. Human plasma was obtained from the Shenzhen Bao’an Authentic TCM Therapy Hospital and was approved by the Shenzhen Bao’an Authentic TCM Therapy Hospital’s Ethics Committee (Shenzhen, China).

### 3.2. Preparation of the Modified Electrode

Firstly, the gold electrode (2 mm in diameter, Gaossunion, Wuhan (China)) was pretreated according to our previous method with little modification [26]. In brief, the gold electrode (GE) was polished with different diameters of γ-alumina polishing powder to obtain the “mirror surface”. The GE was cleaned via ultrasound in deionized water and ethanol for 5 min to remove the residual γ-alumina powder. The GE was then activated with 0.5 M H_2_SO_4_. After that, 8 μL of the MB-P5 treated with TCEP was dropped onto the GE and then placed in the dark overnight at room temperature. Then, the GE was washed with PBS and blown dry with nitrogen. The MCH aqueous solution (1 mM, 8 μL) was dropped onto the GE for 2 h. After washing with PBS and blowing dry with nitrogen, the modified GE was obtained and then stored in a refrigerator at 4 ℃ for later use.

### 3.3. Determination of miR-145 and Clinical Samples

The biotin-modified P1 was equally mixed with P2 and P3 for 15 min. Then, a certain amount of SA-MNPs was added and reacted for 20 min. After magnetic separation, the collected MNPs were washed with PBS and resuspended with PBS. The obtained modified MNPs were mixed with miR-145 and P5. After a period of time, the solution was magnetically separated, and the supernatant was collected. The above-prepared modified GE was dipped into the obtained supernatant and Exo III was also added at the same time. Thirty min later, the obtained GE was washed with PBS and then prepared for electrochemical testing. The electrochemical measurements were carried out on a CHI660D workstation (CH Instruments Inc., Shanghai, China). The current signals before (I_0_) and after (I_1_) treatment of the CSDR/Exo III/miR-145 were obtained separately via different pulse voltammetry (DPV) with the potential voltage range from −0.5 V to 0 V. In DPV, the current value at -0.268V was set as the current signal intensity of electrochemical detection of miR-145. Each experiment was carried out three times. The mean values of three measurements were set as the final signal intensity. Error bars represent the standard deviations of three repetitive experiments. The current change (△i) was obtained based on △i = I_0_ − I_1_. In the specificity experiment, the single-base mismatch (SM), double-base mismatch (DM), triple-base mismatch (TM), and random sequence were detected following the same procedure of detecting miR-145. The collected clinical plasma samples were treated using the miRcute miRNA isolation kit (Tiangen Biotech Co. Ltd., Beijing, China) to obtain the total RNA according to the standard protocol of the instruction of the kits. The total RNA was detected according to the above description procedure for the detection of miR-145.

### 3.4. qRT-PCR Experiment Determination of miR-145

The total RNA from the blood was extracted through an RNeasy midi kit (QIAGEN) according to the manufacturer’s protocol. Then, for the obtained total RNA, a qRT-PCR was carried out via a QIAGEN OneStep RT-PCR kit according to the manufacturer’s protocol. U6 small nuclear RNA (GeneCopoeia) was used as the control. The miRNA was reverse transcribed into cDNA at 50 °C for 30 min and 95 °C for 15 min. The amplification reaction was performed for 40 cycles at 95 °C for 30 s, 55 °C for 30 s, and 70 °C for 30 s. All of the measurements were carried out three times. The miR-145 expression level was normalized to the U6 and evaluated via the 2^−ΔΔCt^ method.

## 4. Conclusions

In summary, we have developed a novel and sensitive electrochemical biosensor for detecting miR-145. This electrochemical biosensor takes full advantage of the synergy effect of CSDR, MNPs, and Exo III. The CSDR is capable of efficiently converting one miR-145 target into the release of numerous pieces of free DNA from the complex of DNA/MNPs. The released DNA can hybridize with the MB-DNA modified on the electrode to form a specific structure, leading to the MB-DNA being digested by Exo III. This developed electrochemical biosensor can distinguish a single-base mismatch from the miR-145. This biosensor has a good linear correlation for detecting miR-145 from 1 × 10^2^ to 1 × 10^6^ aM with a detection limit of 100 aM. In addition, the developed electrochemical biosensor has been successfully applied for the detection of clinical samples. It can distinguish healthy people and stroke patients. The results of this biosensor are also in good accordance with the results of the RT-qPCR. This indicates that the developed electrochemical biosensor has good, potential application for the diagnosis of strokes. Meanwhile, this biosensor can be extended to detect other disease-associated miRNAs or DNAs by adapting the relevant DNA sequences. 

## Data Availability

The data presented in this study are available herein and in the Supporting formations.

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
