# Peer review of "CSDR Coupling with Exo III for Ultrasensitive Electrochemistry Determination of miR-145"

_molecules, 2023, doi:10.3390/molecules28052208_

Round 1
Reviewer 1 Report
Article 1 CSDR coupling with ExoIII for ultrasensitive electrochemistry 2 determination of miR-145
In my opinion, the article is suitable for Molecule Journal, and after a minor revision, it can be accepted in the journal. My suggestions are the following:
1- Reviewing the text shows some writing errors and they should be corrected.
2- The main advantage of the developed method should be highlighted and what is the superiority of the current method over the previous methods?
3- In Table 1, the end of 3 primes has been omitted, it must be corrected. It seems that the sequences are not presented properly.
4- Is the developed method affordable? be included in the results section.
Author Response
Dear Editor:
Thank you for your letter of 19 December and for the reviewer’s comments concerning our manuscript ID“molecules-2099326”Title: CSDR coupling with ExoIII for ultrasensitive electrochemistry determination of miR-145”.
We have found that the comments of editor and reviewer are most insightful and very useful for improving the paper. We have studied their comments carefully and have made corrections which we hope meets their approval. All the changes in the paper are marked in red.
Looking forward to your reply!
Best regards,
Dr. Qinhua Chen
2022.12.25
Comments and Suggestions for Authors
Article 1 CSDR coupling with ExoIII for ultrasensitive electrochemistry 2 determination of miR-145 .In my opinion, the article is suitable for Molecule Journal, and after a minor revision, it can be accepted in the journal. My suggestions are the following:
1-Reviewing the text shows some writing errors and they should be corrected.
Answer:We have carefully revised the relevant writing errors in the manuscript according to the reviewers' comment.
2- The main advantage of the developed method should be highlighted and what is the superiority of the current method over the previous methods?
Answer: we had added the corresponding description in the manuscript according to the reviewers' comment.
Due to the subtly combination of CSDR and Exo III, the developed biosensor exhibited ultra-high sensitivity compared with the previous methods. The powerful separation ability of MNPs can reduce the potential interference of matrix. This can further improve the specificity of the developed electrochemical biosensor.
- In Table 1, the end of 3 primes has been omitted, it must be corrected. It seems that the sequences are not presented properly.
Answer:we had revised the error in the manuscript according to the reviewers' comment.
|
Name |
Sequences description |
|
P1
|
5’ biotin-TTTTTTTTGAA A TG G TG GAA AGG AGG G AG GGA TTC CTG GGA AAA CTGGAC3’ |
|
P2 |
5’ CCTTTCCACCATTTC3’ |
|
P3 |
5’ TTT TCC CAG GAA TCC CTC CCT AAC TGC3’ |
|
P4 |
5’ TT TCC CAG TCC CCT CC TTT CCA CCA TTT3’ |
|
MiR-145 |
5’ G UCCAG UUU UCC CAG GAA UCC CU3’ |
|
SM |
5’ CUCCAG UUU UCC CAG GAA UCC CU3’ |
|
DM |
5’ CACCAG UUU UCC CAG GAA UCC CU3’ |
|
TM |
5’ CAGAG UUU UCC CAG GAA UCC CU3’ |
|
Random |
5’ UUAGCUUAUCAGACUGAUGUUGA3’ |
|
P5 |
5’SH-TTTTTT GTT AGG GAG GGA TTC CTG GGA-MB 3’ |
4- Is the developed method affordable? be included in the results section.
Answer:we had added the corresponding description on page 8 in the manuscript according to the reviewers' comment.
Compared with other related techniques for early diagnosis of stroke, the developed highly sensitive electrochemical biosensor in this study has the advantages of easy to operate, cost-effective, low environmental and equipment requirements, which has the potential of widespread application.
Reviewer 2 Report
Reviewer: In the paper titled “CSDR coupling with ExoIII for ultrasensitive electrochemistry determination of miR-145 " was considered carefully. Although the subject is worthy of investigation, there are some critical technical problems, which make the paper not meet the journal criteria. Thus, this paper may suitable for publication after revision. Some important comments:
1. Table 1 has a formatting error.
2. What is the Linear range of reference 25 in Table 2?
3. Whether the electrochemical immunosensor has specific recognition is not discussed in this paper.
Author Response
Comments and Suggestions for Authors
Reviewer: In the paper titled “CSDR coupling with ExoIII for ultrasensitive electrochemistry determination of miR-145 " was considered carefully. Although the subject is worthy of investigation, there are some critical technical problems, which makes the paper not meet the journal criteria. Thus, this paper may suitable for publication after revision. Some important comments:
- Table 1 has a formatting error.
Answer: we had revised in the manuscript according to the reviewers' comment.
|
Name |
Sequences description |
|
P1
|
5’ biotin-TTTTTTTTGAA A TG G TG GAA AGG AGG G AG GGA TTC CTG GGA AAA CTGGAC3’ |
|
P2 |
5’ CCTTTCCACCATTTC3’ |
|
P3 |
5’ TTT TCC CAG GAA TCC CTC CCT AAC TGC3’ |
|
P4 |
5’ TT TCC CAG TCC CCT CC TTT CCA CCA TTT3’ |
|
MiR-145 |
5’ G UCCAG UUU UCC CAG GAA UCC CU3’ |
|
SM |
5’ CUCCAG UUU UCC CAG GAA UCC CU3’ |
|
DM |
5’ CACCAG UUU UCC CAG GAA UCC CU3’ |
|
TM |
5’ CAGAG UUU UCC CAG GAA UCC CU3’ |
|
Random |
5’ UUAGCUUAUCAGACUGAUGUUGA3’ |
|
P5 |
5’SH-TTTTTT GTT AGG GAG GGA TTC CTG GGA-MB 3’ |
- What is the Linear range of reference 25 in Table 2?
Answer: we had revised in the manuscript according to the reviewers' comment.
|
Analytical Methods
|
Detection Limit |
Linear Range |
Ref. |
|
gold nanoparticle-HCR coupled system |
0.519 nM |
1 pM-1 nM |
22 |
|
fluorescence anisotropy (FA) amplifier Methods |
2.5 nM |
6.0 nM to 21.0 nM |
23 |
|
modified parallel tail-clamps Methods |
2.43nM
|
2 nM to 30 nM
|
24 |
|
eSPR Methods |
0.56 fM |
1.0 fM to 10 nM |
25 |
|
DNA electrochemical aptasensor |
0.27 nM |
2.0 to 80.0 nM |
26 |
|
CSDR coupling with ExoIII |
0.058 fM |
10 to 1 × 10^5 fM |
This method
|
- Whether the electrochemical immunosensor has specific recognition is not discussed in this paper.
Answer:we had revised on page 8 in the manuscript according to the reviewers' comment.
The superior specificity of the electrochemical biosensor for detecting miR-145 is mainly attributed to the synergy effect of the CSDR and MNPs. SM initiation of CSDR is much less efficient than miR-145 because CSDR is specifically triggered by miR-145 in this electrochemical biosensor. In addition, the powerful separation of MNPs can reduce the potential interference of reactants.